# Twenty Years of Experience in Juvenile Nasopharyngeal Angiofibroma (JNA) Preoperative Endovascular Embolization: An Effective Procedure with a Low Complications Rate

**DOI:** 10.3390/jcm10173926

**Published:** 2021-08-31

**Authors:** Andrea Giorgianni, Stefano Molinaro, Edoardo Agosti, Alberto Vito Terrana, Francesco Alberto Vizzari, Alberto Daniele Arosio, Giacomo Pietrobon, Luca Volpi, Mario Turri-Zanoni, Giuseppe Craparo, Filippo Piacentino, Paolo Castelnuovo, Fabio Massimo Baruzzi, Maurizio Bignami

**Affiliations:** 1Neuroradiology Unit, ASST Sette Laghi-Circolo Hospital, 21100 Varese, Italy; andrea.giorgianni@asst-settelaghi.it (A.G.); albertovito.terrana@asst-settelaghi.it (A.V.T.); francescoalberto.vizzari@asst-settelaghi.it (F.A.V.); fabio.baruzzi@asst-settelaghi.it (F.M.B.); 2Department of Biotechnology and Life Sciences, Division of Neurosurgery, University of Insubria, 21100 Varese, Italy; edoardo_agosti@libero.it; 3Department of Biotechnology and Life Sciences, Division of Otorhinolaryngology, University of Insubria, 21100 Varese, Italy; albertodaniele.arosio@gmail.com (A.D.A.); tzm@inwind.it (M.T.-Z.); paolo.castelnuovo@me.com (P.C.); 4Department of Surgical Specialities, Division of Otorhinolaryngology, ASST Sette Laghi-Circolo Hospital, 21100 Varese, Italy; 5Department of Head and Neck Surgery and Otorhinolaryngology, European Institute of Oncology IRCCS, 20122 Milano, Italy; giacomo.pietrobon@gmail.com; 6Department of Otorhinolaryngology, ASST Lariana, University of Insubria, 22100 Como, Italy; luca.volpi81@gmail.com (L.V.); bignami67@me.com (M.B.); 7Department of Surgery, ASST Lariana, University of Insubria, 22100 Como, Italy; 8Head and Neck Surgery & Forensic Dissection Research Center (HNS&FDRc), Department of Biotechnology and Life Sciences, University of Insubria, 21100 Varese, Italy; 9Diagnostic and Interventional Neuroradiology Unit, ARNAS Civic Hospital, 90127 Palermo, Italy; gcraparo@yahoo.it; 10Radiology Unit, ASST Sette Laghi-Circolo Hospital, 21100 Varese, Italy; filippo.piacentino@asst-settelaghi.it

**Keywords:** JNA, embolization, interventional neuroradiology, HNS

## Abstract

Juvenile nasopharyngeal angiofibroma (JNA) is a benign tumor of the nasal cavity that predominantly affects young boys. Surgical removal remains the gold standard for the management of this disease. Preoperative intra-arterial embolization (PIAE) is useful for reductions in intraoperative blood loss and surgical complications. In our series of 79 patients who underwent preoperative embolization from 1999 to 2020, demographics, procedural aspects, surgical management and follow-up outcome were analyzed. Embolization was performed in a similar fashion for all patients, with a superselective microcatheterization of external carotid artery (ECA) feeders and an injection of polyvinyl alcohol (PVA) particles, followed, in some cases, by the deployment of coils . Procedural success was reached in 100% of cases, with no complications such as bleeding or thromboembolic occlusion, and surgical intraoperative blood loss was significantly decreased. In conclusion, PIAE is a safe and effective technique in JNA treatment, minimizing intraoperative bleeding.

## 1. Introduction

Juvenile nasopharyngeal angiofibroma (JNA) is a highly vascularized and histologically benign tumor of the nasal cavity and paranasal sinuses, with aggressive behavior and locally invasive growth patterns [1]. It comprises 0.05% of head and neck tumors and predominantly occurs in young boys, with a mean age of presentation of 15 years [2,3]. The best treatment to date remains surgical removal of the tumor [3]. Preoperative embolization is used for virtually all cases of JNA, resulting in reduction in intraoperative bleeding, occlusion of surgically inaccessible arterial feeding vessels, decreased operative time and improved surgical visualization, identification and protection of adjacent structures [4,5,6]. This results in a significant reduction in overall surgical complications and, despite some reports of safe resection without embolization, it is considered to be the standard of care in most centers [7,8,9]. In this study, we describe our single-center experience in preoperative JNA devascularization with the injection of polyvinyl alcohol (PVA) into major lesion feeders, highlighting the safety and efficacy of this technique. Furthermore, we put emphasis on the detection of external carotid artery (ECA)-internal carotid artery (ICA) anastomoses, defining the main red flags to be considered during preoperative intra-arterial embolization to avoid intraprocedural iatrogenic embolic complications.

## 2. Materials and Methods

### 2.1. Data Collection

The study was performed in compliance with the Helsinki Declaration and with policies approved by the Insubria Board of Ethics. All patients involved in the study signed a consent form to publish their clinical photographs whenever useful.

We performed a retrospective analysis of 79 patients treated surgically at our Institution for JNA between 1999 and 2020 who underwent PIAE of ECA branches with the sole usage of PVA. Angiographic patterns, Radkowski stage [10], surgical approach, surgical time, blood loss, age and follow-up imaging were also listed in the database in Appendix A. CT and MRI scans were performed in all patients in order to assess Radkowski stage (Figure 1).

The main outcomes considered were the incidence of complications related to embolization and/or surgery, residual disease rate and intraoperative blood loss. 

### 2.2. Endovascular Embolization

The same approach was performed for every patient, with right groin puncture and placement of 6F femoral sheath, catheterization of internal/external carotid artery (ICA/ECA) and vertebral artery (VA) with angiographic study of their vascular regions (Figure 2), followed by 6F guide catheter (Envoy MPC 90 cm, Cordis) in proximal ECA and superselective catheterization of lesion feeders. Microcatheters used (Rebar 18, Medtronic; SL-10, Stryker) ranged from 0.0165 in to 0.021 in of internal diameters; guidewires used (Traxcess 14, Synchro 10, Synchro 14, Stryker) ranged from 0.010 in to 0.014 in. A control run was then performed from the microcatheter to look for dangerous collaterals and determine the precise position of the distal tip (Figure 3). Embolization was then performed using PVA particles (Contour, Boston Scientific, Marlborough, MA, USA) with different sizes—ranging from 250–355 µm to 500–710 µm—in a slow infusion using blank roadmap visualization to achieve as proper distal penetration as anatomically possible until complete stasis of flow within each feeding vessel was achieved. Adjunctive coil embolization with GDC platinum coil was performed if particle embolization turned out to be incomplete, especially in the case of hypertrophied IMA. At the end of the procedure, control angiography was performed from both ICA and ECA to assess the percentage of tumor feeders embolized. Successful embolization was determined as a lack of contrast in the vascular territory of the embolized vessel (Figure 4).

## 3. Results

In total, 79 patients were included in this series. The mean age was 18 years (range 10–63 years); all of them were male (100%). The most common symptom was epistaxis (55%), followed by nasal obstruction (50%). According to the classification of Radkowski et al., 3/79 (3.8%) type IA, 7/79 (8.9%) type IB, 26/79 (32.9%) type IIA, 7/79 (8.9%) type IIB, 21/79 (26.6%) type IIIA and 15/79 (18.9%) type IIIB tumors were treated. PIAE with PVA intra-arterial injection was performed in all patients. All cases displayed tumor arterial supply from ECA and/or ICA circulations on 2D angiograms, with a total number of arterial tumor feeders embolized in a given session ranging between 1 and 5. 

The technical success of angiography and embolization of almost one big feeder was 100%. Embolization of the JNAs was performed in all cases (79/79) (100%); from distal sphenopalatine branches of the internal maxillary artery in 35/79 cases (44.3%); from ascending pharyngeal artery branches in 20/79 cases (25.3%); from an accessory branch of the middle meningeal artery (MMA) in 7/79 cases (8.9%); from the facial artery and a deep temporal branch of the MMA in 5/79 cases (0.6%); and, in 64/79 cases (81%), procedures were performed under general anesthesia, while the other 15 (18.9%) were performed under conscious sedation.

There was no post-procedural bleeding and there were no thrombo-embolic cerebral ischemic complications in any patient. In no case were there any complications such as vascular dissections, groin hematomas or other complications related to vascular microcatheterization or embolization. Neck pain was experienced by a few patients, and was promptly resolved with analgesic medications. Tumors was removed in all cases within 24 h after embolization. All patients underwent surgery through an endoscopic endonasal approach. All patients were neurologically intact after surgery. Diagnosis of JNA was confirmed histopathologically after surgery. 

Follow-up imaging was predominantly performed with MRI. Residual lesions were identified in 7/78 patients (8.9%). In post-surgical remnant JNA patients, the mean size of the preoperative lesion and the presence of vascular afferent from the ICA was greater than in JNAs, in which gross total resection occurred. Of all the post-surgical remnant JNAs, only 2/79 (2.5%) underwent new surgical treatment. Demographic, clinical and surgical data of the 79 patients are summarized in Table 1.

## 4. Discussion

In this study, we documented an excellent safety profile of PIAE with PVA, reporting no complications directly related to the embolization.

JNA is a rare, benign, vascular lesion of the skull base that affects young adolescent males most commonly between 9 and 19 years of age [3,10]. It is highly aggressive and associated with significant morbidity. Its tendency for skull base erosion, intracranial extension (20% of cases) and high vascularity (vascular component in a fibrous stroma with single endothelial lining) make surgical resection challenging, with a relevant risk for blood loss during resection, post-surgical remnants and lesion recurrence [9]. 

JNA commonly originates in the posterolateral wall of the nasal cavity, near the superior margin of the sphenopalatine foramen, with progressive diffusion to the anterior nasal cavity, maxillary sinus, pterygoid region, infratemporal fossa and middle cranial fossa [11,12,13]. Signs and symptoms are most often related to tumor extension into the nose, leading to nasal obstruction and epistaxis [10,13,14]. Feeding vessels usually arise from the external carotid system via the internal maxillary artery or ascending pharyngeal artery, but can be highly variable, often with heterogeneous vascularization patterns originating from contralateral ECA, petrous and cavernous branches of ICA, such as mandibulo-vidian artery, inferolateral trunk and ECA-ICA anastomosis, such as ethmoid branches of the ophthalmic artery, which are often related to bigger dimensions [15,16,17]. 

Traditionally, the open transfacial approach has been the gold standard for JNA excision [1]. In recent years, the advent of endonasal endoscopic approaches (EEAs) has revolutionized the surgical management of these lesions, reducing JNA post-surgical morbidity and recurrence rates [14]. The main advantages of the endoscopic endonasal route are better magnification of the lesion, the dissection of the surgical planes between the lesion and healthy tissue and better cosmetic outcomes. However, JNA resection can still be complicated by massive hemorrhaging because of a rich vascular supply [14,15]. 

In order to reduce intraoperative bleeding, facilitate surgical lesion removal and improve a patient’s post-operative course, over time, preoperative embolization techniques have been established [18]. The main techniques used for preoperative JNA embolization are endovascular arterial catheterization and direct percutaneous puncture [16,17,18,19].

Pharmacological treatments also have been described to minimize the intraoperative bleeding. Thakar et al. described a significant difference between prepubertal and postpubertal patients in their response to flutamide. Indeed, in postpubertal patients, 6 weeks preoperative may lead to partial tumor regression, facilitating surgical excision and limit morbidity [20].

PIAE is the current most accepted treatment for JNA, minimizing intraoperative bleeding and reducing surgical morbidity [15,17,21,22]. This technique not only reduces the blood supply to the lesion, but the diagnostic preoperative digital subtraction angiography (DSA) highlights the JNA specific vascularization patterns, guiding the surgeon to plan the approach and to delineate lesion areas of increased bleeding risk [15,23]. However, the intra-arterial embolization has some technical limitations, mainly due to the presence of non-embolizable small feeders and to the vascular spasm caused by catheter endovascular manipulation [19]. Furthermore, the presence of ECA-ICA anastomosis directly involved in the vascular supply limits complete JNA devascularization for the risk of inadvertent injection of embolic material into ICA circulation by anterograde crossing from ECA branches through the tumor feeders [16,23,24]. These embolic complications can lead to retinal and cerebral strokes, with iatrogenic blindness and permanent brain damage [24,25,26].

The widely used standard approach for JNA is embolization with particles such as PVA, embospheres (Guerbet Biomedical, Louvres, France) and gelfoam (Upjohn Co., Kalamazoo, MI, USA), all of which have been used successfully for the PIAE of head/neck tumors, as well as in the central nervous system [15,22]. The use of liquid embolic agents (e.g., Onyx), also by percutaneous direct puncture, has been reported to allow for a deeper penetration to tumor capillaries with improved fluoroscopic visibility, as well as a lower risk of catheter adherence [17]. When using PVA, because of its irregular profile (“flakes”), a minimum particle size of more than 150 μm, with a range from 150 to 350 μm, is believed to provide the best compromise between safety (collaterals) and efficient devascularization. As only a temporary occlusion can be achieved, an interval no longer than 7 days between particle embolization and surgery is essential to ensure sufficient devascularization [17,19,27].

## 5. Conclusions

In this retrospective analysis, PIAE has demonstrated itself to be a safe technique (absence of major intra- or periprocedural hemorrhagic or ischemic complications) and, above all, effective in reducing intraoperative bleeding. Additionally, offering improved intraoperative visibility also reduces postoperative JNA residual rates.

## Figures and Tables

**Figure 1 jcm-10-03926-f001:**
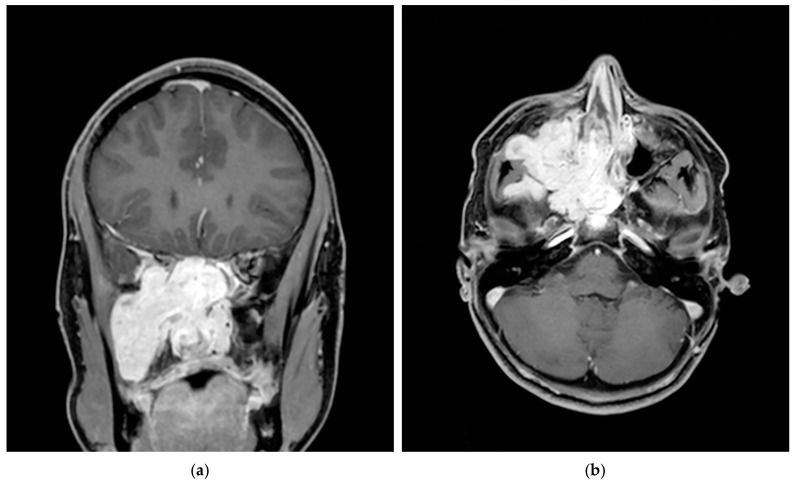
Preprocedural MRI scan (Gd-enhanced T1 Gradient-Echo 3D): coronal (**a**) and axial (**b**) views showing large JNA of right nasopharyngeal mass with expansion of the pterygopalatine fossa and extension into the infratemporal fossa.

**Figure 2 jcm-10-03926-f002:**
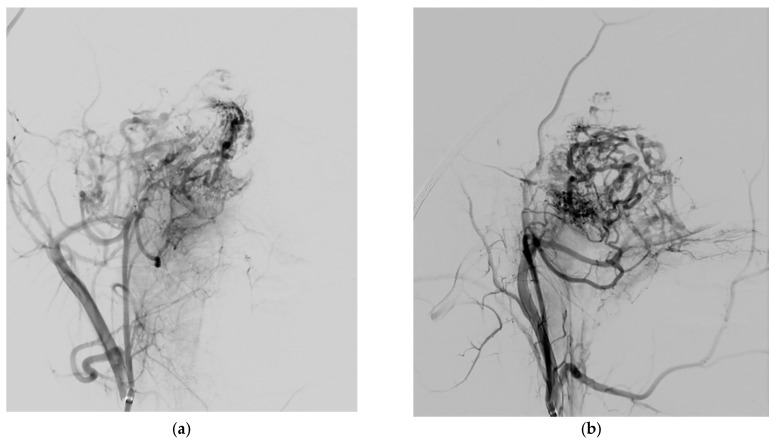
Preoperative DSA: selective catheterization of proximal ECA. Posteroanterior (PA) (**a**) and laterolateral (LL) (**b**) views of the JNA with major feeders from sphenopalatine branches of the distal internal maxillary artery (IMA) and from the ascending pharyngeal artery (APhA).

**Figure 3 jcm-10-03926-f003:**
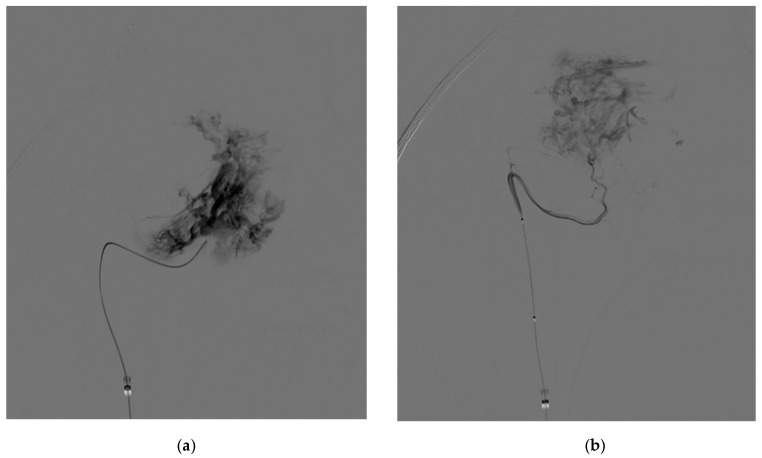
Intraprocedural DSA: PA view of superselective injection of distal IMA (**a**) and APhA (**b**) feeders.

**Figure 4 jcm-10-03926-f004:**
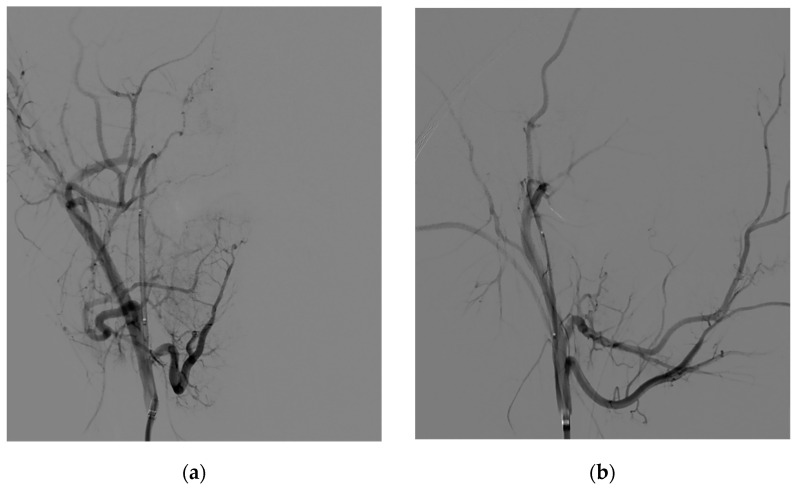
Postprocedural DSA: PA (**a**) and LL (**b**) views of the JNA showing successful embolization of the lesion.

**Table 1 jcm-10-03926-t001:** Demographic, clinical and procedural characteristics of the 79 patients.

Variables		Data
Age (years)	Mean	18
	Median	20
	Range	10–63
Symptoms	Epistaxis	55%
	Nasal obstruction	50%
	Rhinolalia	14%
	Headache	12%
	Proptosis	10%
	Diplopia	6%
	Decreased visual acuity	2%
Radkowski classification	Type IA	4%
	Type IB	9%
	Type IIA	33%
	Type IIB	9%
	Type IIIA	26%
	Type IIIB	19%
Intraoperative blood loss (mL)	Mean	784
	Range	40–5200
Surgical time (min)	Mean	217
	Range	95–625
Neuroimaging follow up (months)	Mean	25
	Median	12
	Range	1–127

## Data Availability

Data are available on request due to restrictions, e.g., privacy or ethical reasons.

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
