# Peer review of "Twenty Years of Experience in Juvenile Nasopharyngeal Angiofibroma (JNA) Preoperative Endovascular Embolization: An Effective Procedure with a Low Complications Rate"

_jcm, 2021, doi:10.3390/jcm10173926_

Round 1

Reviewer 1 Report

The  is really interesting and well written. They report the 20-years experience in the preoperative endovascular embolization treatment of the juvenile nasopharyngeal angiofibroma (JNA). The authors performed a superselective microcatheterization of external carotid artery (ECA) feeders through injection of polyvinyl alcohol (PVA) particles. The structure is correct; however, some minor corrections are required to improve the overall quality: 

Abstract

  • please remove the text highilighted ''morbidity, and reducing postoperative residual rate. and Keywords: JNA; embolization

Methods correct and clear description

Results

  • please specify the number of each features described. Only percentage is present sometimes while conversely only numbers sometimes. Report always number (percentage)

Discussion

  • Several preoperative treatment have been described to minimize the bleeding intraoperative. Thakar et al. described significant difference between Prepubertal and postpubertal patients in their response to flutamide. Indeed, in postpubertal patients, 6 weeks preoperative may lead to partial tumor regression, facilitating surgical excision and limit morbidity. please cite the manuscript doi:10.1002/hed.21667

Reviewer 2 Report

Many thanks for the opportunity to review this paper. I found this work very interesting, well structured, and comprehensive. I only suggest specifying the abbreviations throughout the text, and also in the chronological order of appearance (e.g., ECA-ICA at the end of the Introduction section and at the beginning of the “Endovascular embolization” paragraph should be reversed). Furthermore, in the Table 1 “Surgical time” should probably be reported in “min” and not “ml”.

Author Response

Many thanks. Please see the attachment below
